# Brain Activity Recognition Method Based on Attention-Based RNN Mode

## Song Zhou and Tianhan Gao *

Software College, Northeastern University, Shenyang 110169, China; 1710467@stu.neu.edu.cn
* Correspondence: gaoth@mail.neu.edu.cn

**Abstract:** Brain activity recognition based on electroencephalography (EEG) marks a major research orientation in intelligent medicine, especially in human intention prediction, human–computer control and neurological diagnosis. The literature research mainly focuses on the recognition of single-person binary brain activity, which is limited in the more extensive and complex scenarios. Therefore, brain activity recognition in multiperson and multi-objective scenarios has aroused increasingly more attention. Another challenge is the reduction of recognition accuracy caused by the interface of external noise as well as EEG's low signal-to-noise ratio. In addition, traditional EEG feature analysis proves to be time-intensive and it relies heavily on mature experience. The paper proposes a novel EEG recognition method to address the above issues. The basic feature of EEG is first analyzed according to the band of EEG. The attention-based RNN model is then adopted to eliminate the interference to achieve the purpose of automatic recognition of the original EEG signal. Finally, we evaluate the proposed method with public and local data sets of EEG and perform lots of tests to investigate how factors affect the results of recognition. As shown by the test results, compared with some typical EEG recognition methods, the proposed method owns better recognition accuracy and suitability in multi-objective task scenarios.

**Keywords:** brain activity recognition; EEG; attention-based RNN model; XGBoot classifier; brain–computer interface

## 1. Introduction

Brain activity recognition has been one of the key topics in the field of brain science research in recent years. It has great potential to change traditional brain science applications, such as diagnosis of diseases related to the nervous system, auxiliary applications for the disabled and the elderly, device control and disease monitoring, etc. For people with limited behavioral ability, brain activity recognition can help them complete basic daily activities and enable them to live more independently. At the same time, with the continuous development and maturity of deep learning technology, EEG-based brain activity recognition is more accurate, which effectively improves the use value of brain activity recognition technology in the field of actual diagnosis and treatment and assisted living [1]. Brain activity is mostly represented by electroencephalogram (EEG) signals. Voltage fluctuations of brain neurons are recorded in a noninvasive manner by placing electrodes on the scalp [2]. EEG records the changes of radio waves during brain activity, which is the overall reflection of the electrophysiological activities of brain nerve cells on the cerebral cortex or scalp surface. EEG is distributed in different frequency bands according to different brain activities, so it can grasp the current state of brain activity.

In recent years, researchers have conducted in-depth research on brain activity recognition, but it still faces several challenges such as multiperson and multitask classification. First, some researches focus on EEG feature representation, and a series of EEG feature recognition methods are proposed. For instance ,Vzard et al. [3] adopted common spatial patterns (CSP) and LDA for the preprocessing of EEG data, concluding 71.59% accuracy

rate under binary alert state [4]. Multiclass CSP (mCSP) and Self-Regulated Interval Type-2 Neuro-Fuzzy Inference System (SRIT2NFIS) classifiers were exploited for the recognition of four classes of EEG-based motor imagery and achieved the accuracy of 54.63%. Shiratori et al. [5] combined mCSP and random forest for three-class EEG-based moving image tasks, achieving a 56.7% accuracy rate. Through the analysis of these studies, it is found that the accuracy of the classification and recognition algorithm is not enough to support the application standard of EEG under actual demand. This is mainly because the complexity and high dimensionality of the EEG signal make the feature representation of the original EEG data have a great influence on the classification accuracy. On the basis of summarizing these studies, finding a high-accuracy EEG data classification algorithm to support the application of EEG in real life is one of the important problems we need to solve. Second, most of the current applications that use EEG recognition are for disease diagnosis (such as epilepsy and Alzheimer's disease), which only requires binary classification (normal or abnormal). However, emerging application fields mostly rely on multiclass EEG classification. Regarding this [6], adopted SVM to classify the four types of EEG data with a classification accuracy of 70%. Vernon et al. [7] proposed a set of EEG classification methods based on CNN for different BCI paradigms, which is robust enough in practice. In addition, the above research mainly focuses on EEG data recognition in single-person with single-task mode. As for the multiperson and multitask scenarios, Ridha Djemal et al. [8] employed an LDA (linear discriminant analysis) classifier to classify two datasets with nine and three subjects, while the classification accuracy is still unsatisfactory.

Based on the above discussion, a novel EEG data recognition method is proposed. The correlation coefficient matrix is calculated in multiperson and multitask scenarios by analyzing the similarity of EEG signals. EEG signal features are then extracted through the attention-based RNN model. Finally, the extracted features are fed into XGBoost classifier for identifying EEG data type, with every type matched with given brain activities. Here are the contributions of the paper:

(1) The paper proposes an EEG analysis-based brain activity recognition approach able to be directly applied in primitive EEG data analysis, thereby reducing dependence on professional EEG knowledge.

(2) The attention-based RNN model is adopted to improve the effect of EEG classification through adjusting hidden layer neuron's size.

(3) The dedicated experiments are conducted on EEG datasets comprising 560,000 samples under 20 topics and 5 categories to evaluate this proposed method. According to the comparison with other prime EEG classification approaches, the method in this paper can improve the accuracy by about 10%.

The subsequent parts of the research are presented in the following order: Section 2 briefs on relevant technical preliminaries. The proposed brain activity recognition method is elaborated in Section 3. The experiments and comparative analysis are given is Section 4. Finally, the paper is concluded in Section 5.

## 2. Preliminaries

### 2.1. EEG Analysis

As indicated by EEG data analysis, there are five frequency patterns, namely, delta, theta, alpha, beta and gamma patterns, in line with the correlation of human behavior states [9–11]. Every decomposed EEG pattern includes signals associated with certain brain information. Each EEG pattern represents an active pattern of a specific brain state and qualitative evaluation of awareness as below.

Delta pattern (0.5–4 Hz) is related to deep sleep, which indicates the low awareness of subjects. Theta pattern (4–8 Hz) embodied in light sleep is in the state of low awareness.

Alpha mode (8–12 Hz) is mostly seen in closed eyes and deep relaxation, when the subject is in a state of medium consciousness.

Beta pattern (12–30 Hz) is the dominant type, where the subject continues to widen the horizon and improve awareness. The majority of human daily activities, like eating, walking, and talking belong to this pattern.

Gamma pattern (30–100 Hz) suggests the co-cooperation of few brain areas for performing certain motor and cognitive functions. It is related to stronger awareness.

We judge there exists an internal relationship between EEG pattern and awareness degree [10,12]. Due to the growth of band frequency (Delta pattern, Theta pattern, Alpha pattern, Beta pattern, Gamma pattern), awareness degree continually improves. Aforementioned arguments could be deduced from the two factors as below: Firstly, EEG pattern is associated with brain neuron activities. Essentially, EEG signal is measured with voltage fluctuations caused by ion currents in the activity of brain neurons [13]. Secondly, there exists a relationship between awareness degree and brain neuron activities. On an intuitive level, those subjects who have stronger awareness will have more activated neurons.

For identification techniques, EEG signal feature is expected to be steady and distinguishable. The steady feature suggests fine system robustness, meaning the system can still recognize users even they are experiencing trivial fluctuations psychologically or physically, like fatigue. The distinguishable feature indicates the varying reactions of different subjects to EEG signals [14,15]. According to the analysis results, the paper takes a Delta pattern to analyze the state of the subjects, which contains the most stable and unique information of the user's brain activity state recognition.

### 2.2. Attention-Based RNN Model

After EEG pattern decomposition, constructed Delta pattern is input into the attention-based Encoder-Decoder RNN structure [16] for acquiring more representative features in brain activity recognition. The attention-based model is a method to free the encoder-decoder structure from a fixed-length internal representation, by maintaining the encoder pair input in the RNN model. The attention-based system enables the RNN model to allocate weight to various parts at input layer, thereby providing a new solution for clarifying the correspondence between the input layer and the output layer. Attention-based RNN models have been widely used in speech recognition, NLP (natural language processing), as well as computer copyright management [17].

### 2.3. XGBoot

XGBoost is a gradient boosting algorithm with a residual decision tree. The basic idea is to add a decision tree to the model one by one. When a decision tree is added, the overall effect of the model (the objective function is reduced) is improved . The XGBoot classifier combines a set of classification and regression trees (CART) and attempts to make use of the information from the input data as much detail as possible [18]. Multiple trees are built with their own leaves and corresponding score. Compared with the gradient boosting algorithm, XGBoost proposes a more regularized model formalization to prevent overfitting, with the engineering goal of pushing the limit of computation resources for boosted tree algorithms to achieve better performance. Here, XGBoost uses the idea of random forest for reference and adds a regular term to the cost function to control the complexity of the model. The regular term includes the number of leaf nodes of the tree and the square sum of L2 modules of score output on each leaf node. From the perspective of bias variance trade-off, the regular term reduces the variance of the model and makes the learned model simpler, so as to prevent overfitting.

### 2.4. Brain–Computer Interface (BCI)

BCI refers to a communication control system which can directly connect with external devices without relying on nerves and muscles around the brain. BCI technology involves the cross fusion of multiple disciplines such as electronic signal detection, signal processing, and pattern recognition [11]. At present, the main purpose of BCI is to provide a new way of communication with the outside world for people with normal thinking but with

impaired physical function. The mainstream research direction is to obtain the patient's EEG and control the external equipment, so as to replace the impaired physical function of the patient, or to assist the patient's missing function [19].

## 3. The Proposed Method

In the proposed method, subjects' brain wave information is gathered from portable brain wave equipment. Subjects are requested to stay in the relaxing environment to ensure stability and reliability of the EEG signal. First, the collected EEG samples should receive preprocessing for erasing DC offset impact on normalizing. $\delta$-band data are then divided from preprocessed data with EEG mode decomposition. Moreover, $\delta$-band data are fed into attention-based RNN model for classification. Finally, the current state of subjects is recognized according to the classification result. Table 1 lists the parameters involved in this paper.

**Table 1.** Notation.

| Parameter | Explanation |
|:---:|:---:|
| E | EEG raw data |
| $E'$ | Preprocessed EEG data |
| $X^i$ | Data in the $i$-th layer in attention-based RNN |
| I | The number of layers in attention-based RNN |
| $N^i$ | The number of dimensions of $X^i$ |
| Y | The one-hot label of emotion |
| $Y'$ | The predicted status form attention-based RNN |
| K | The number of categories of subject status |
| $f(\bullet)$ | The linear function |
| C | The intermediate code |
| $T(\bullet)$ | The output calculation procedure of LSTM cell |
| $T'(\bullet)$ | The final hidden state calculation procedure of LSTM cell |
| $W'_a$ | The unnormalized attention weights |
| $W_a$ | The normalized attention weights |
| $C_a$ | The attention-based intermediate code |
| $X_D$ | The feature of deep learning |
| $S_D$ | The final status of subject |

### 3.1. EEG Signal Preprocessing

Primitive EEG samples are preprocessed for erasing DC offset. Different EEG acquisition devices will introduce constant noise components when recording signals. Thus, in the preprocessing stage, a constant DC offset should be subtracted from primitive signal E. Data normalization generates a significant impact on data processing at each unit and feature scale.Normalization is a linear operation, which makes EEG data more in line with independent and identical distribution conditions, and reduces the offset caused by internal corvariate shift, so that the input data of (0, 1) standard normal distribution is converted into slightly deviated (0, 1) standard normal distribution data. This operation can ensure that the input data basically maintain a normal state, and the input is nonlinear, which keeps the data away from the saturation region of the activation function and speeds up. Z-score normalization will be used in this paper. Preprocessed brain wave data E' is computed as below(1),

$$E' = \frac{(E - DC) - \mu}{\sigma} \tag{1}$$

where *DC* is direct current noise offset, $\mu$ suggests *E-DC* average, and $\sigma$ represents standard deviation of *E-DC*.

The EEG signal belonging to the delta band (0.5–4 Hz) is the most stable and accurate for emotion recognition. In order to separate the signals in the delta band, we use a filter, in which the order is set to 3, the low cut-off is 0.5 Hz and the high cut-off is 4 Hz. All

dimensions of the preprocessed data are sent to the band-pass filter in turn, and finally the decomposed delta mode is obtained $\delta$.

Since human consciousness is naturally related to the mental and physical state of each person, and the delta band contains the most stable EEG consciousness record, the features here refer to the data frequency corresponding to the EEG signal in the delta band under different reactions.

### 3.2. Attention-Based RNN

The decomposed EEG data are inputted into an RNN model that introduces an attention mechanism to obtain more information about the subject's brain activity. The general encoder-decoder RNN ignores the importance of the feature dimension to the output sequence and defaults to the weight of entire feature dimensions of input sequence. Feature scales of EEG data are matched with the EEG device's sensor nodes. In order to assign different weights to different dimensions of EEG data, the attention mechanism is introduced into an encoder-decoder RNN model, which is composed of three parts: encoder module, attention mechanism module and decoder module. The encoder module should compress input delta-band data into over-encoded C. The attention module enables the encoder to compute a better intermediate code $C_a$ through creating weight W at various dimensions. The decoder module illustrates attention-based code $C_a$ into an EEG recognizable signal.

Suppose that the data at the *i*-th layer are stated as $X^i = X^i_j; i \in [1, 2, \ldots, I], j \in [1, 2, \ldots, N^j]$ in which is denoted the *j*-th dimensional data of $X^i$, I represents the number of neural network layers, and $N_i$ represents the number of dimensions in $X^i$. . At the first layer, in case of $=\delta$, input sequence contains $\delta$-band data. Output sequence as represents the subject's current status, in which K is the number of subject state category. Operation $f(\bullet)$ is defined as below:

$$f(X^i) = X^i * W + b \tag{2}$$

$$f(x^{i-1}_j, X^i_{j-1}) = X^{i-1}_j * W' + X^{i-1}_j * W'' + b' \tag{3}$$

where $W, b, W', W''$ denote the corresponding weights and deviation parameters. The encoder contains several nonrecursive fully connected neural network layers and one Long Short-Term memory (LSTM) layer. The nonrecursive layer is used to construct and fit a nonlinear function to filter the input $\delta$-band data. The data for non-recursive layers can be calculated as (4),

$$X^i + 1 = f(X^i) \tag{4}$$

The LSTM layer compresses the output of the nonrecurrent layer into a fixed-length sequence as the transfer code C. Assuming that the LSTM model is located in the $i'$-th layer of the encoder, the code C is the output of the LSTM model, $C = X^{i'}_j$, and $X^{i'}_j$ can be measured by

$$X^{i'}_j = T(c^{i'}_{j-1}, X^{i-1}_j, X^{i'}_{j-1}) \tag{5}$$

where $c^{i'}_{j-1}$ denotes the hidden state of the $(j-1)$-th LSTM cell, the operation $T(\bullet)$ denotes the calculation of LSTM, which can be inferred from the following equations:

$$X^{i'}_j = f_o \odot tanh(c^{i'}_j) \tag{6}$$

$$c^{i'}_j = f_f \odot c^i_{j-1}{}' + f_i \odot f_m \tag{7}$$

$$f_o = sigmoid(f(X^{i'-1}_j, X^{i'}_{j-1})) \tag{8}$$

$$f_f = sigmoid(f(X^{i'-1}_j, X^{i'}_{j-1})) \tag{9}$$

$$f_i = sigmoid(f(X^{i'-1}_j, X^{i'}_{j-1})) \tag{10}$$

$$f_m = tanh(f(X^{i'-1}_j, X^{i'}_{j-1})) \tag{11}$$

where represent the output gate, forget gate, input gate and input modulation gate, respectively, and $\odot$ denotes the element multiplication. The attention-based module receives the state of the hidden layer as a nonstandardized attention weight , which can be measured by

$$W_a{}' = T'(c_{j-1}^{i'}, X_j^{i-1}, X_{j-1}^{i'}) \tag{12}$$

$W_a'$ is further normalized to obtain

$$W_a = softmax(W_a') \tag{13}$$

The softmax function normalizes attention weights within [0, 1]. Consequently, attention weight is considered as the probability estimate investigating if there is a correlation between code C and output. In attention mechanism, connection code C will be converted to $C_a$ after being weighted.

$$Y' = X^I = T(C_a) \tag{14}$$

Both connection code C and standardized attention weight can be used in real-time training. Decoder is used to receive attention-based code C for predicting the subject's present status. denotes the predicted status trained by attention-based RNN model.

Ultimately, crossover function is adopted to compute the predicted cost of $Y'$ and real status of Y. L2-norm (with parameter $\lambda$) is selected for preventing overfitting. Iteration threshold of attention-based RNN is set to be $n_{iter}$. The Weighted code Ca is in a linear correlation with output layer and prediction result. Suppose the model is well-trained and low-cost, weighted coding can be considered a high-quality manifestation of user brain activity information. Depth feature $X_D$ of RNN model is Ca which can be mapped as ultimate subject brain activity.

### 3.3. Brain Activity Recognition

Extreme XGBoot is used in learned depth feature $X_D$ classification for determining the present brain activity status of subjects. $X_D$ is taken for training classification regression tree and predicting subject information. Suppose there is m CART, and $x_d \in X_D$ refers to a single sample of depth feature, ultimate recognition results are:

$$y_m = l(x_d) \tag{15}$$

$$S_D = L(\sum_1^M y_m), m = 1, 2, \ldots, M \tag{16}$$

where $l$ is a single tree's classification function, $y_m$ is the $m$-th tree's predicted status, $L$ is the mapping from single tree prediction space to final prediction space, and $S_D$ reflects ultimate recognition results.

## 4. Experiments and Analysis

In this section, we evaluate the feasibility and accuracy of the proposed method on a common EEG dataset.

### 4.1. Datasets and Tasks

PhysioNet eegmmidb (EEG motor movement/imagery database) database, as a common EEG database, has been used for the assessment of proposed method. Data collection is finished with BCI 2000 system, including 64 channels and a 160 Hz sampling rate. During data collection, various categories of test scenarios are played in a cycle to record the corresponding EEG data of subjects. Each test corresponds to a task as below.

Task 1 (Relaxed): In cases in which the screen exhibits blue sky and white clouds, subjects imagine they are relaxing in a quiet grassland.

Task 2 (Excited): In cases in which the scenario is switched to the concert, subjects imagine they are at their favorite concert in an excited state.

Task 3 (Tired): In cases when the scenario is switched to the sports field, subjects imagine they are in exhausted and performing strenuous exercises.

Task 4 (Anxiety): When the screen shows a test, subjects imaging they are in an anxious state in an exam.

Task 5 (Feared): When a horror image is exhibited on the screen, subjects imagine they are staying in the terrible environment.

560,000 EEG samples (28,000 samples per subject) are selected from 20 subjects for follow-up experiments. Every sample is one vector comprising 64 elements in 64 channels, and corresponds to one of the above tasks, which are marked as a class (from 0 to 4).

*4.2. Evaluation*

The recognition in the experiment is a two-point problem that includes the four cases in Table 2.

**Table 2.** Different cases of recognition.

| Instance | Predict Positive | Predict Negative |
|---|---|---|
| Positive class | True Positive (TP) | True Negative (TN) |
| Negative class | False Positive (FP) | False Negative (FN) |

Based on the above cases, we define the following criteria.

(1) *Accuracy*: the proportion of all correctly predicted samples. Accuracy is a measure of how good a model is

$$Accuracy = \frac{TP + TN}{FP + FN + TP + TN} \tag{17}$$

(2) *Precision*: the proportion of all positive samples with the correct number of positive samples divided by the classifier.

$$Precision = \frac{TP}{FP + TP} \tag{18}$$

(3) *Recall*: the proportion of all correct results being accurately predicted.

$$Recall = \frac{TP}{FN + TP} \tag{19}$$

(4) *F1 Score*: a weighted average of the model accuracy and recall rate. Its maximum value is 1, and the minimum value is 0.

$$F1\ Score = 2 * \frac{precision * recall}{precision + recall} \tag{20}$$

(5) ROC (Receiver Operating Characteristic): the relationship between TPR (True Positive Rate) and FPR (False Positive Rate) at various threshold settings.

(6) AUC (Area Under the Curve): the area under the ROC curve. The value of AUC drops in the range [0.5, 1].

*4.3. Experimental Results*

4.3.1. Parameter Settings

In this experiment, the attention-based RNN model is trained with training data, and test data are input into model for feature extraction through the XGBoost classifier. In this attention-based RNN model, the encoder is composed of one input layer (64 nodes), three non-recursive fully connected hidden layers (64 nodes) and one recursive LSTM layer (121 units). This decoder comprises one fully connected hidden layer (121 nodes) and one output layer (64 nodes). The learning rate is 0.001. L-2 norm parameter is set as 0.001.

This training dataset includes 20 batches in a size of 28,000, with the number of training iterations set to 2000. With the XGBoost classifier, the learning rate $\eta$ is 0.7, the parameter associated with minimum loss reduction and number of leaves $\gamma$ is 0, the maximum depth of the tree is 6, the subsampling rate of the training instance is 0.9 for overfitting prevention.

### 4.3.2. Results

560,000 EEG samples were gathered for the experiment, each containing a 64-dimensional feature vector and a factual correctness label. The raw EEG data are randomly assigned to form a training data set (532,000 samples) and a test data set (28,000 samples) after normalization by the z-score approach. The attention-based RNN model extracts representative features from 121 hidden neurons and inputs them into XGBoot classifier. Table 3 presents the results' confusion matrix data.

**Table 3.** Different cases of recognition.

| | class | 0 | 1 | 2 | 3 | 4 | Accuracy | Recall | F1 Score | AUC |
|---|---|---|---|---|---|---|---|---|---|---|
| | | **Ground Truth** | | | | | **Evaluation** | | | |
| | 0 | 3753 | 0 | 312 | 238 | 420 | 0.8036 | 0.7759 | 0.7908 | 0.9434 |
| Predict | 1 | 392 | 7903 | 518 | 448 | 492 | 0.8058 | 0.9352 | 0.8532 | 0.9532 |
| Label | 2 | 238 | 178 | 3931 | 334 | 209 | 0.8103 | 0.7637 | 0.7724 | 0.9502 |
| | 3 | 132 | 126 | 207 | 3326 | 162 | 0.8356 | 0.7301 | 0.7795 | 0.9496 |
| | 4 | 357 | 359 | 238 | 206 | 3402 | 0.7386 | 0.7402 | 0.7458 | 0.9408 |
| average | | | | | | | 0.7987 | 0.789 | 0.7883 | 0.9474 |

The average accuracy, recall, F1 score and AUC of 28,000 data (5 task categories from 20 subjects) are 0.7987, 0.7890, 0.7883 and 0.94574, respectively. It can be seen that the recall, F1 score and AUC of class1 are the highest, which means that class1 features the most obvious difference and is the easiest to recognize. On the contrary, the sample of class4 is the most difficult to recognize. The same conclusion can be obtained by the ROC curve as shown in Figure 1. The diagonal lines represent random classifiers in which TPR = FPR. The classifier will have better performance as long as theROC curve is closer to the upper left corner. At the same time, the AUC of all categories of samples is above 0.94, indicating the stability and high efficiency of chosen classifier.

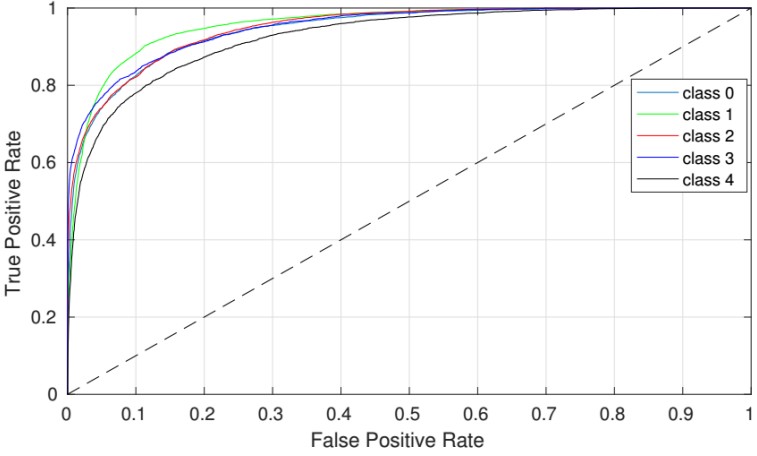

**Figure 1.** ROC curves for 5-class classification by XGBoost.

### 4.3.3. Accuracy Analysis

The experiments corresponding to the 5 tasks are repeated 5 times, with training data ratios of 60%, 70%, 80%, 90% and 95%. As shown in Figure 2, the training time is proportional to the amount of training data, and the training time is acceptable when the

test error reaches the lowest point of 0.206. It can be concluded that the larger the training data, the higher the test accuracy.

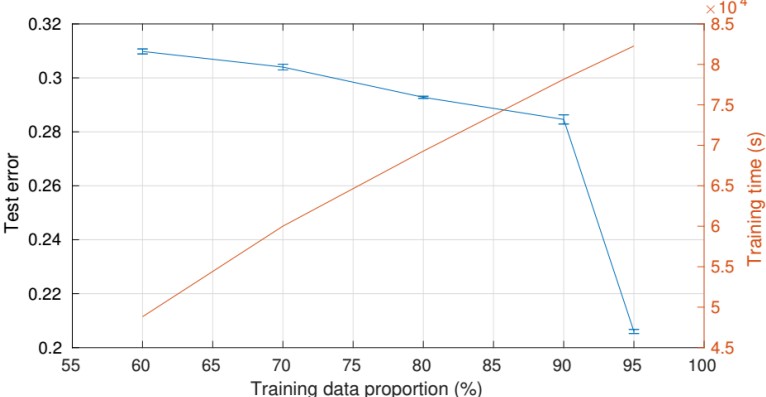

**Figure 2.** The relationship between the test error and error bars, training time, as well as training data proportion.

Figure 3 shows the iterative relationship between the test error and the various training data ratios. All curves tend to converge after 1600 iterations, further demonstrating that a greater test data ratio may effectively reduce the test error.

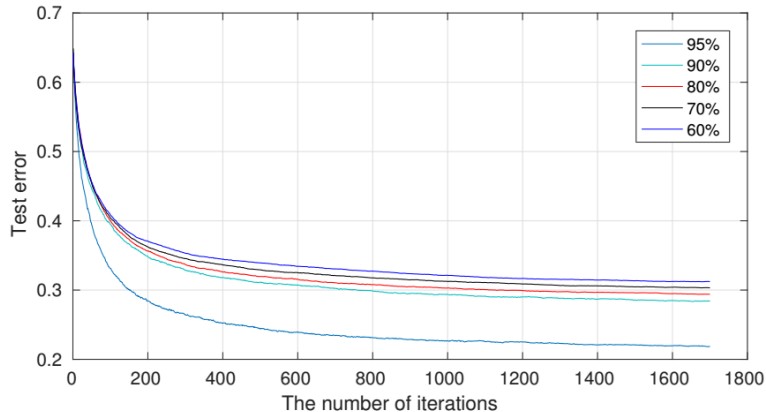

**Figure 3.** The relationship between the test error and error bars, training time, as well as training data proportion.

The number of neurons in the hidden layer of the attention-based RNN model represents the extracted feature dimension, which has a great influence on the quality of the feature extraction and the classification result. In order to analyze the relationship between the number of neurons and the classification effect, we set the number of designed neurons in the range from 30 to 200.

We can see from Figure 4 that in the first stage (0–120), the test error decreases as the number of neurons increases. While in the second stage (greater than 120), the test error is around 0.21, and it slightly fluctuates. At the same time, there appears to be a linear relationship between the training time curve and the number of neurons. Although the difference between the test error curve and the training time curve reaches a minimum of about 100 neurons, the test error is still large. The test error reaches the minimum number of neurons, 121, and the training time is acceptable.

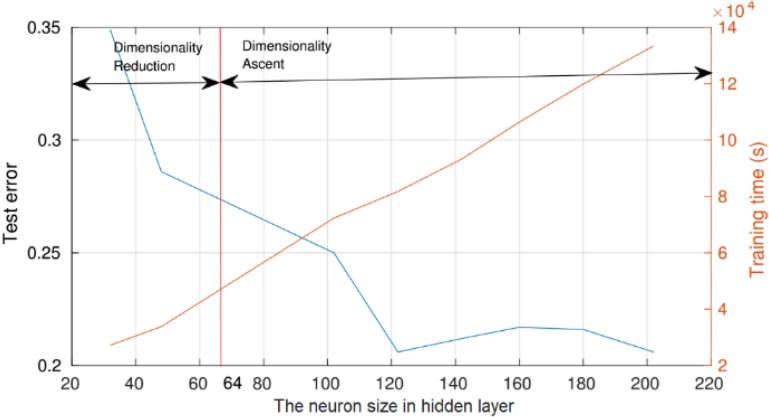

**Figure 4.** The relationship between the test error and error bars, training time, as well as training data proportion.

### 4.3.4. Comparative Analysis

XGBoost is adopted to classify the fine EEG features generated by the attention-based RNN model. The proposed method(RNN+AB+XGBoost) is compared with several widely used classification methods on the same EEG dataset to test the effectiveness.The comparison of various classification methods is shown in table 4.

**Table 4.** Comparison of various classification methods. AB denotes attention-based method; RF denotes Random Forest; DT denotes Decision Tree; EIG denotes the eigenvector-based dimensionality reduction method used in eigenface recognition; PCA denotes Principal Component Analysis; DWT denotes Discrete Wavelet Transform.

| No | 1 | 2 | 3 | 4 | 5 | 6 | 7 |
|---|---|---|---|---|---|---|---|
| Classifier | SVM | RNN | XGBoost | CNN | RF | DT | LDA |
| Acc | 0.355 | 0.674 | 0.6322 | 0.524 | 0.661 | 0.306 | 0.342 |

| No | 8 | 9 | 10 | 11 | 12 | 13 | 14 |
|---|---|---|---|---|---|---|---|
| Classifier | RNN+SVM | PCA+XGBoost | PCA+RNN+XGBoost | EIG+RNN+XGBoost | EIG+PCA+XGBoost | DWT+XGBoost | RNN+AB+XGBoost |
| Acc | 0.689 | 0.723 | 0.643 | 0.501 | 0.63 | 0.67 | 0.804 |

Firstly, we use 7 common classifiers (SVM, RF, DT, LDA, XGBoot, RNN and CNN) to classify the data and evaluate which classifier is most suitable for the original EEG data. Then, 4 types of feature extraction methods (PCA, AE, EIG and DWT) are tested to explore the most suitable EEG feature representation method. The experimental results show that, in the single-person and single-objective mode, the basic recognition and prediction effect can be guaranteed through the classifiers such as RNN, LDA and XGBoot. While the XGBoot classifier can achieve classification accuracy of 0.689 without preprocessing and feature extraction of EEG raw data. However, when facing the multiperson and multi-objective scenario, the experiments show that the accuracy of RNN+AB+XGBoost can reach 0.804, which is better than the other models.

Compared with the existing algorithms, our algorithm can initially meet the application of real medical scenarios in the recognition accuracy of brain emotional state with the needs of multiscene and multitask conditions. At the same time, the algorithm can effectively meet the integration of overall and local features of EEG signals, and is more suitable for the BCI system. In the next step, we will improve the classification performance by establishing a multiview model for multiple types of EEG signals, especially by establishing multiple models, each of which processes one class. Meanwhile, other deep learning algorithms such as DNN have been successfully applied to EEG based BCI, but these methods are mainly applied to a single BCI paradigm. As the focus of future research, we try to design a CNN architecture to accurately classify EEG signals from different BCI paradigms. In addition, it is also a direction for follow-up research to learn

from the algorithm of deep learning in image and language recognition processing and apply it to EEG signal processing.

## 5. Conclusions

This paper focuses on the classification and recognition of EEG data under the condition of multiperson and multi-objective scenarios to find the differences between different brain wave modes. The normalized method is adopted to preprocess the raw EEG data and then be inputted into the attention-based RNN model for training. The model is tested with five different scenarios. Finally, the validity and efficiency of the proposed method are evaluated on the EEG datasets containing 560,000 samples (belonging to five categories). As part of future research, we will build different view models of multiclass EEG to further improve the classification effect.

**Author Contributions:** Conceptualization, S.Z. and T.G.; methodology, S.Z.; software, S.Z.; validation, S.Z. and T.G.; formal analysis, S.Z.; data curation,S.Z.; writing—original draft preparation, S.Z.; writing—review and editing, S.Z. and T.G.; All authors have read and agreed to the published version of the manuscript.

**Funding:** Supported by the Fundamental Research Funds for the Central Universities under Grant Number: N2017004.

**Institutional Review Board Statement:** Ethical review and approval were waived for this study, due to this paper only uses machine learning technology to analyze and study EEG data, and does not involve human ethics, so the ethical approval is not necessary in this paper.

**Informed Consent Statement:** Not applicable

**Data Availability Statement:** Data sharing not applicable.

**Conflicts of Interest:** The authors declare no conflict of interest.

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
