# Peer review of "Brain Activity Recognition Method Based on Attention-Based RNN Mode"

_applsci, doi:10.3390/app112110425_

Round 1

Reviewer 1 Report

The paper presents a novel method for classification of EEG signals based on attention-RNN networks. The  proposed method used the delta band of EEG signals and obtained higher accuracy in comparison to other methods. 

In general, I consider the machine learning will help to move the filed of neuroscience forward  and I think this paper may add something into the existing literature. My main concerns are related to the clarity in which the text is written which in many places is very hard to follow. 

  • Intro-> The intro is really hard to follow and needs a lot of reogarnisation. The first paragraph introduces EEG and concludes with the application of deep learning. Then it moves to EEG again and the third paragraph goes to machine learning. I think everything can be more clearly aligned to help the reader following the text. I think that the machine learning part of the intro also need some work. I think that in general it is not so important the classification rates that those algorithms obtained but the what those methods lack and what is the problem you are trying to solve. I lack that information in the introduction.
  • Methods: the first and second part of the EEG preliminaries is repeated. After the description of the EEG bands you start suggesting that :

    We judge there exists an internal relationship between EEG pattern and awareness degree". I think this is something well documented in the literature so maybe add some refs in there or rephrase that statement. The last paragraph of that sentence is really hard to follow so please review it. Finally, there is a sentence at the end saying "This is an example of a quote" -> I believe that has to be removed.

    Give indications on how you decompose the EEG signal Pre-processing can affect different aspects of the signal. Filters, ICA yes or not...plese describe in detail. Also when you talk about the features extracted for the algorithm, what features were extracted in detail?

In the XGBoot section you mention that when a decision tree is added it improves the performance. Improves how? In the same section you mention a formalisation method of regularisation. Which method? Please describe.

Also please describe why normalisation affects may have an impact on the data processing at each unit (158). 

  • Results: Im the evaluation results I believe there is a typo in the accuracy formulation formula. In the numerator should be TN + TP right? You wrote FN + TP. In line 248 you wrote "It can be concluded that the larger the test data, ...." I believe it should be the larger the training data, ...
  • Discussion -> I know that this not mandatory given the journal guidelines but neuroscientists (for instance) are in general not experts in machine learning and they would benefit from a discussion of the presented results. What makes the algorithm you present better in comparison to existing ones? How can even be improved more? How would perform against more advances methods like DNN or Reservoir Computing methods? I miss all this in the text either together with the results or in a small section discussing the results. I believe that is a very important step to help the readers.
  • I always ask that in all the papers I review. Is the code available in any repository? If not why?
  • Finally, in general I think the text needs to be proof-reader. I'm not a native speaker either but many sentences are hard to understand and there are some typos. Some examples:
    • Line 76 sention4 should be section 4. 
    • Line 101 A for mentioned should be together aforementioned.
    • Line 236 calss should be class.

Reviewer 2 Report

This work by Zhou et. al. demonstrates a brain activity recognition method using the attention-based RNN. This is a good case of the application of AI to the medical area. The results of the method are compared with that of other classifiers, which demonstrate good performance. The manuscript is well organized and the results are clearly presented. I only have minor concerns about this manuscript that the authors should consider addressing.

The source of the public data. The source of the public data was labeled as reference 19 in the text but was not added to the reference list.

The authors should consider making their code available to the readers if there is no conflict of interest to it, which will benefit the research community of this area.

Round 2

Reviewer 1 Report

Thank you very much for addressing all my comments and adding all the edits to the text. In general, some English style still needs to be improved and some further proofreading is needed (lots of full stops misplaced, words stick together...) but I recommend the text to be accepted for publication.